# Enhanced Simultaneous Machine Translation with Word-level Policies

**Kang Kim**\*, **Hankyu Cho**\*

XL8 Inc.

{kai, matthew}@xl8.ai

## Abstract

Recent years have seen remarkable advances in the field of Simultaneous Machine Translation (SiMT) due to the introduction of innovative policies that dictate whether to READ or WRITE at each step of the translation process. However, a common assumption in many existing studies is that operations are carried out at the subword level, even though the standard unit for input and output in most practical scenarios is typically at the word level. This paper demonstrates that policies devised and validated at the subword level are surpassed by those operating at the word level, which process multiple subwords to form a complete word in a single step. Additionally, we suggest a method to boost SiMT models using language models (LMs), wherein the proposed word-level policy plays a vital role in addressing the subword disparity between LMs and SiMT models. Code is available at https://github.com/xl8-ai/WordSiMT.

## 1 Introduction

Simultaneous Machine Translation (SiMT) commences the translation process while simultaneously receiving the input, making it an effective approach for applications that require minimal latency such as simultaneous interpretation or live broadcast. The development of a novel policy is central to research efforts in SiMT. This policy dictates the translation process by determining whether to execute a READ or WRITE action at each step of the process.

Neural SiMT models, like offline Neural Machine Translation (NMT) models, commonly employ Byte Pair Encoding (BPE) (Sennrich et al., 2016) or similar techniques to encode an input sentence into a sequence of tokens. Typically, a single READ or WRITE action of a SiMT policy is responsible for handling an encoded token, which may sometimes be a word but often a subword.

---

\*Equal contribution.

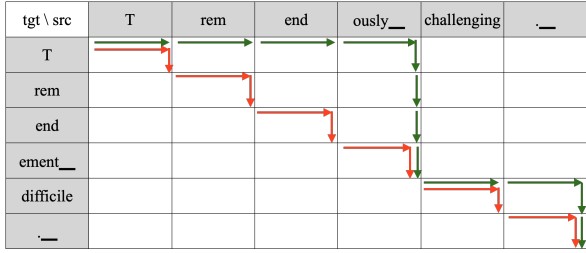

Figure 1: Illustration of both token-level Wait-1 (red arrow lines) and word-level Wait-1 (green arrow lines) policies. The symbol "＿" at the end of a token indicates a word boundary.

The development of BPE-based SiMT models and their policies has resulted in researchers focusing on working at the subword level. The performance analysis and implementation of many SiMT systems, to the best of our knowledge, have been carried out on encoded sequences of source and target subwords, rather than on the original source and target sentences[1]. This has led to two critical issues that need to be addressed.

The first issue is the lack of a standardized tokenization and encoding scheme, meaning that different implementations may employ varying token sequences to encode identical text. This variability can impact latency evaluation results and complicate score comparisons across different systems.

The second issue is the missed opportunity to process more source tokens before writing each target token without added latency. For a BPE-based SiMT model, the input must be received on a word-by-word basis to ensure proper encoding of each word into a sequence of subwords. Consequently, when the model encodes a word and performs a READ to process only a subword, it delays the reading of the remaining subwords without any benefit in actual latency[2], and may adversely impact

---

[1]We provide a list of reference works pertaining to this case in A.1

[2]Here we assume that the processing time of a READ is negligible, compared to the interval between receiving words.

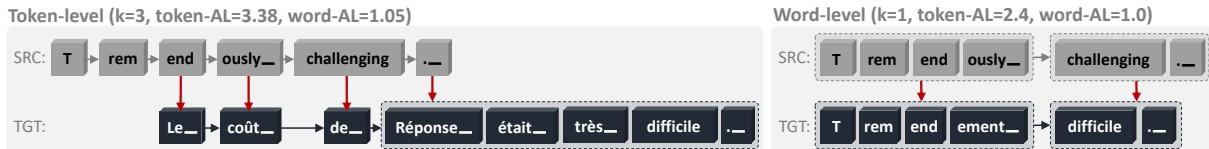

Figure 2: An exemplary case depicting the difference between token-level and word-level Wait-k policies and their Average Lagging (AL) scores. The token-level model begins translating in the middle of reading "Tremendously" and fails to recover from the incorrectly translated target prefix. On the other hand, the word-level model processes "Tremendously" in a single READ action and produces a correct translation.

translation quality by causing the model to rely on incomplete representations extracted from partially-read words. Similarly, performing a WRITE to generate a subword earlier than the remaining subwords does not necessarily reduce latency, as the subword must wait for a complete word to be generated before it can be displayed.

In this paper, we show that establishing the unit of measuring and operating SiMT systems at the word level, rather than the subword level, offers a viable solution to these issues. Specifically, to tackle the first issue, we propose word-level latency metric calculation for measuring the latency of SiMT systems. This not only enables consistent comparisons between different systems but also provides a more accurate reflection of the actual latency experienced in SiMT applications that display translation results word by word.

To address the second issue, we illustrate that an existing token-level policy can be transformed into a word-level policy that inherently overcomes the second issue, resulting in improved performance. Word-level policies take into consideration word boundaries and perform a READ or WRITE action to sequentially process a sequence of tokens that form a complete word. This conversion process can be applied to any token-level policies, and our experiments reveal that state-of-the-art fixed and adaptive policies exhibit significantly better performance when transformed into their word-level counterparts. Notably, these word-level policies often outperform token-level policies, even when evaluated using a token-level latency metric, due to their enhanced utilization of input and output tokens.

Additionally, to boost translation accuracy, we suggest incorporating a pre-trained language model (LM) into a SiMT model, where the word-level policy plays a crucial role as a pivotal component. One of the major hurdles in utilizing an LM for SiMT is the vocabulary mismatch between the SiMT model

and the LM. The difficulty of handling subword disparities when utilizing an LM for a downstream task is widely acknowledged (Liu et al., 2021a; Wang et al., 2022), and it becomes particularly problematic in SiMT, as inconsistent subwords between the LM and SiMT model make processing the same source or target prefix challenging. Our study demonstrates that our proposed word-level policy effectively tackles this challenge, enabling a successful integration of LMs into SiMT systems.

## 2 Related Work

### 2.1 Simultaneous Machine Translation

SiMT systems that employ a fixed policy utilize a pre-defined sequence of READ and WRITE operations for each source sentence. STATIC-RW (Dalvi et al., 2018) and Wait-k (Ma et al., 2019a) policies first read k source tokens, then alternate between reading and writing a single token. Elbayad et al. (2020) propose the multi-path training of a Wait-k model to train a single model that supports different k values at test time. Zhang et al. (2021) improve Wait-k policy using knowledge distillation from an offline MT model, while Zhang and Feng (2021) suggest a Mixture-of-Experts Wait-k Policy where predictions from multiple k values are combined inside a single model.

In contrast, research efforts on adaptive policies focus on the development of dynamic decision-making processes for READ/WRITE actions. Cho and Esipova (2016) firstly introduce model-based adaptive criteria for Neural SiMT. Gu et al. (2017) propose to learn a policy by using reinforcement learning. Raffel et al. (2017) introduce Monotonic Attention that ensures monotonic alignment in the attention mechanism. Succeeding works improve it by extending the alignment window (Chiu and Raffel, 2017; Arivazhagan et al., 2019), extending it as monotonic multi-head attention (MMA) (Ma et al., 2019b) or learning transposed policies between the forward and backward models. (Zhang and Feng,

2022c). Zheng et al. (2020) derive a policy by composing Wait-k models trained with different values of k. Zhang and Feng (2022a) model to predict the alignment between each target token and the source token. Zhang and Feng (2022b) measure accumulated information from source tokens to decide whether to write the current target token.

Despite significant advancements, the impact of operating policies at the word level has not been thoroughly explored in existing works, which have mainly focused on developing and evaluating systems at the token level. In this paper, we address this gap by demonstrating that implementing various types of policies at the word level consistently outperforms their token-level counterparts.

## 2.2 Utilizing pre-trained LM for MT

Since the successful utilization of Transformer-based LMs pre-trained on large text corpora for downstream NLP tasks (Devlin et al., 2019; Liu et al., 2019; Lample and Conneau, 2019), the utilization of these models for MT has become a significant research area. Several studies have demonstrated the effectiveness of incorporating encoder-only LMs into NMT models. Weng et al. (2020); Yang et al. (2020) combine the LM representations with the encoder's representation using a gating mechanism. Zhu et al. (2020) propose attention between BERT and both the encoder and decoder. Weng et al. (2022) leverage mBERT as an encoder and introduce a decoder that attends to grouped representations of the encoder output.

Another research direction focuses on developing LMs with the encoder-decoder architecture designed for NMT as the target downstream task (Lewis et al., 2020; Liu et al., 2020). These models show improvements particularly for low-resource language pairs. To enhance their adaptation for MT, various methods have been proposed, including fine-tuning specific parts of the LMs (Cooper Stickland et al., 2021), reducing domain mismatch and overestimation (Wang et al., 2022) and mitigating the copying behavior (Liu et al., 2021b).

The integration of pre-trained LMs into SiMT remains an underexplored area of research. To date, Indurthi et al. (2022) is the only related study we are aware of. It improves MMA by integrating the LM's prediction of the next target token, which is encoded using their model's vocabulary before being inputted into the model. However, this approach sacrifices the semantic coherence of the original tokens due to token fragmentation. Additionally, their approach falls under target-side LM integration, overlooking the potential advantages of source-side LM integration.

In this paper, we demonstrate an effective way of integrating a source-side LM into SiMT systems, offering a more versatile solution that can be integrated into most existing neural SiMT models. Building upon previous research conducted in offline MT (Zhu et al., 2020), we introduce essential modifications, with a particular focus on word-level policies as a pivotal component. The effective management of vocabulary mismatches between the LM and the SiMT model is contingent upon the successful implementation of a word-level SiMT policy, a key aspect that we address in our study.

## 3 Proposed Methods

In this section, we propose the concept of employing a word-level latency metric and outline our conversion process for translating token-level policies into their word-level equivalents. Additionally, we present an integration of LM into SiMT, highlighting the advantages of utilizing word-level policies.

### 3.1 Preliminaries

Given a source sentence $\mathbf{x} = (x_1, x_2, ...x_n)$, the goal of a SiMT model is to generate a target sentence of $\mathbf{y} = (y_1, y_2, ...y_m)$ while minimizing latency metrics. A SiMT model's policy, represented by the variable $g_i$, determines the number of source tokens to process before predicting target token $y_i$. Then the probability of generating $\mathbf{y}$ given $\mathbf{x}$ is formulated as follows:

$$p(\mathbf{y}|\mathbf{x}) = \prod_i^{|\mathbf{y}|} p(y_i|\mathbf{x}_{\leq g_i}, \mathbf{y}_{<i}; \theta) \quad (1)$$

where $\theta$ is the model's parameters which are commonly optimized with a cross-entropy loss.

Transformer encoder-decoder model (Vaswani et al., 2017) is currently the most widely used architecture for SiMT. To avoid redundant encoding of the input sequence after each READ operation, the encoder is typically modified to encode the source tokens unidirectionally (Elbayad et al., 2020). Alternatively, more advanced techniques like the recurrent Linear Transformer (Kahardipraja et al., 2021) or Partial Bidirectional Encoding (Iranzo Sanchez et al., 2022) can be adopted to enhance the encoding capabilities further.

During the evaluation of SiMT systems, translation quality is commonly assessed in conjunction with the latency required for generating translations. Various metrics have been proposed to calculate latency scores, with Average Lagging (AL) (Ma et al., 2019a) being the most commonly used metric.

## 3.2 Measuring latency based on the word level

As detailed in A.1, a substantial body of prior research work assesses the performance of SiMT systems by utilizing a latency metric on encoded source tokens under different tokenization and encoding schemes. This practice results in each system being evaluated on non-identical token sequences for the same dataset, thereby making it challenging to accurately compare scores across different systems.

To address this, we propose word-level latency score calculation by considering the word boundaries in token-level sequences. Specifically, when the first token of a source word is processed through a READ operation, we consider it as reading the corresponding word. Similarly, when the last token of a target word is written via a WRITE operation, we consider it as writing that word. By doing so, the latency scores are calculated consistently, regardless of the tokenization and encoding of the input. This ensures that results from different systems can be compared fairly.

## 3.3 Word-level SiMT policies

The proposed word-level policy restricts a SiMT policy's transition from READ to WRITE or vice versa to occur exclusively at the boundaries of words. Any token-level policy can be transformed to operate at the word-level by following the conversion process we outline below.

Concretely, we ensure that a word-level policy does not write a target token in the middle of reading a sequence of source tokens that make up a word. To accomplish this word-level READ, we delay $g_i$ until it reaches the nearest source word boundary. We formally define $r_i$ that has a refined value of $g_i$ based on the word boundaries in $\mathbf{x}$ as follows:

$$r_i := \min\{j|j \geq g_i \wedge j \in B_S\} \qquad (2)$$

Here, $B_S$ denotes the indices of the source words' last tokens. Substituting $r_i$ for $g_i$ as a policy transforms it into another policy that upholds the same

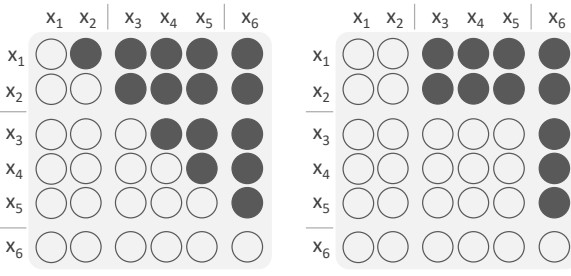

Token-level unidirectional encoding    Intra-word bidirectional encoding

Figure 3: Comparison of masking in the token-level unidirectional attention (left) and the intra-word bidirectional encoding (right). Word boundaries are represented by vertical/horizontal bars on each axis.

decision-making criterion while ensuring uninterrupted reading of an entire word when initiating the reading of a token.

Similarly to the word-level READ, we design a word-level WRITE to balance READ and WRITE actions throughout the translation. To achieve this, we can modify $r_i$ such that it writes until it produces a token that ends with an end-of-word symbol. We define $w_i$ that satisfies this as follows:

$$b_i := \min\{j|j \geq i \wedge j \in B_T\} \qquad (3)$$

$$w_i := \begin{cases} r_i, & \text{if } i = 1 \vee b_{i-1} \neq b_i \\ w_{i-1}, & \text{otherwise} \end{cases} \qquad (4)$$

where $B_T$ denotes the indices of the target words' last tokens and $b_i$ represents the index of the final token in the word that includes $y_i$. By employing $w_i$ in place of $r_i$ (or $g_i$), we ensure that the policy consistently composes entire words without interruptions from any READ actions. This approach effectively reduces latency by facilitating faster writing of certain tokens compared to the original policy, thereby compensating for the increased latency resulting from word-level READ operations. Figure 1 provides a visual comparison between word-level and token-level policies in the context of Wait-1, with the word-level policy encompassing both word-level READ and WRITE operations.

## 3.4 Intra-word bidirectional encoding

Unidirectional encoding in SiMT is vital for managing computational complexity and training efficiency. However, it has an inevitable consequence of weakening the source sequence representations compared to bidirectional encoding. This is an additional factor contributing to the lower translation

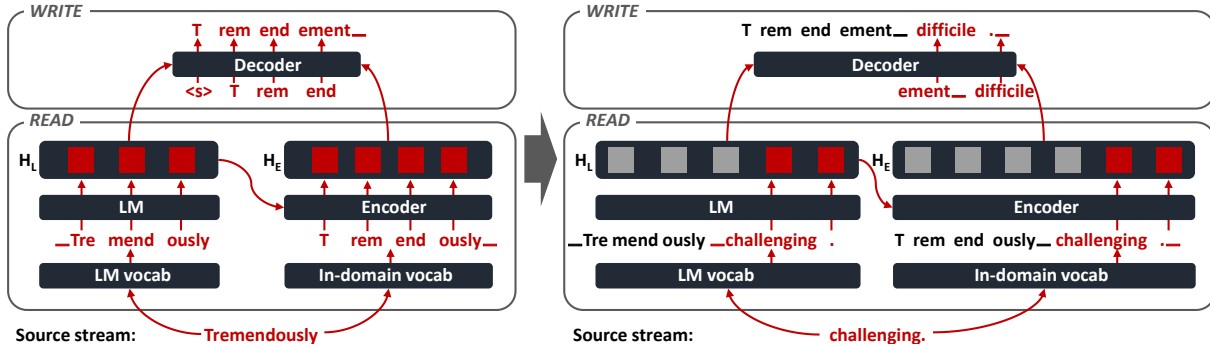

Figure 4: Illustration of LM-fused attention with the word-level Wait-1. Word and model activations processed at a specific stage are highlighted in red. When a source word is received, it is independently encoded by the LM's vocabulary and the in-domain vocabulary. The hidden activations of the word from the LM is then utilized in the encoder. The decoder generates a sequence of tokens for a word by using both the LM and encoder activations.

quality of SiMT compared to offline models, along with the early translation from partial inputs.

To mitigate this issue, we utilize a technique called *intra-word bidirectional encoding*. At the word level, this approach involves unidirectional encoding for each word in the input sentence, meaning past words cannot attend to future words. However, at the subword level, subwords within the same word can attend to each other, allowing past subwords to attend to future subwords within the same word. Since READ operates at the word level in word-level policies, this encoding does not require recomputation during each WRITE operation. It only necessitates a single forward pass, similar to token-level unidirectional encoding. However, it can produce a better encoded representation by enabling attention to more tokens An example masking to enable intra-word bidirectional encoding is depicted in Figure 3.

### 3.5 Integration of LM into SiMT through word-level policies

In this subsection, we showcase an additional benefit of word-level policies when integrating an LM into a SiMT system. One of the key challenges in this integration is the vocabulary mismatch between the LM and the SiMT model, which hinders ensuring that both models process an equal amount of input prefix at each translation step.

One possible solution is to use the LM's vocabulary for the SiMT model. However, the LM's training data may not align well with the specific domain targeted by the SiMT system (Wang et al., 2022). This can result in suboptimal vocabulary for the SiMT model compared to a vocabulary obtained from in-domain data (Liu et al., 2021a). An-

other option is to explore methods to bridge vocabulary gaps (Kim et al., 2019; Sato et al., 2020; Liu et al., 2021a), but they are either validated only in certain transfer learning scenarios or require an additional training phase to train adapters or fine-tuning the entire LM using pre-training data.

In this paper, we introduce a method for leveraging an LM in a manner that facilitates the integration of an off-the-shelf LM into a SiMT model, utilizing a word-level policy, regardless of vocabulary mismatches and the internal structure of the SiMT model. Specifically, we employ an LM fused attention for both the encoder and decoder, following the approach outlined in (Zhu et al., 2020), but with two notable modifications.

Firstly, we replace BERT with a decoder-only auto-regressive LM (Radford et al., 2019; Lin et al., 2022) for unidirectional encoding of the input, aligning with SiMT models for efficient training and inference. Secondly, the attention between the SiMT model and the LM occurs when both models execute a word-level READ for an input word. This ensures they interact only when they process an equal amount of input prefix, naturally resolving the synchronization issue. Additionally, as they align at every word boundary, the SiMT model can operate independently with a vocabulary derived from in-domain data, while the LM continues to use its original vocabulary. Unlike methods targeting specific SiMT models (Indurthi et al., 2022), our approach can benefit any Neural SiMT model with any decoder-only LM. Figure 4 illustrates the proposed integration of the LM with word-level Wait-1.

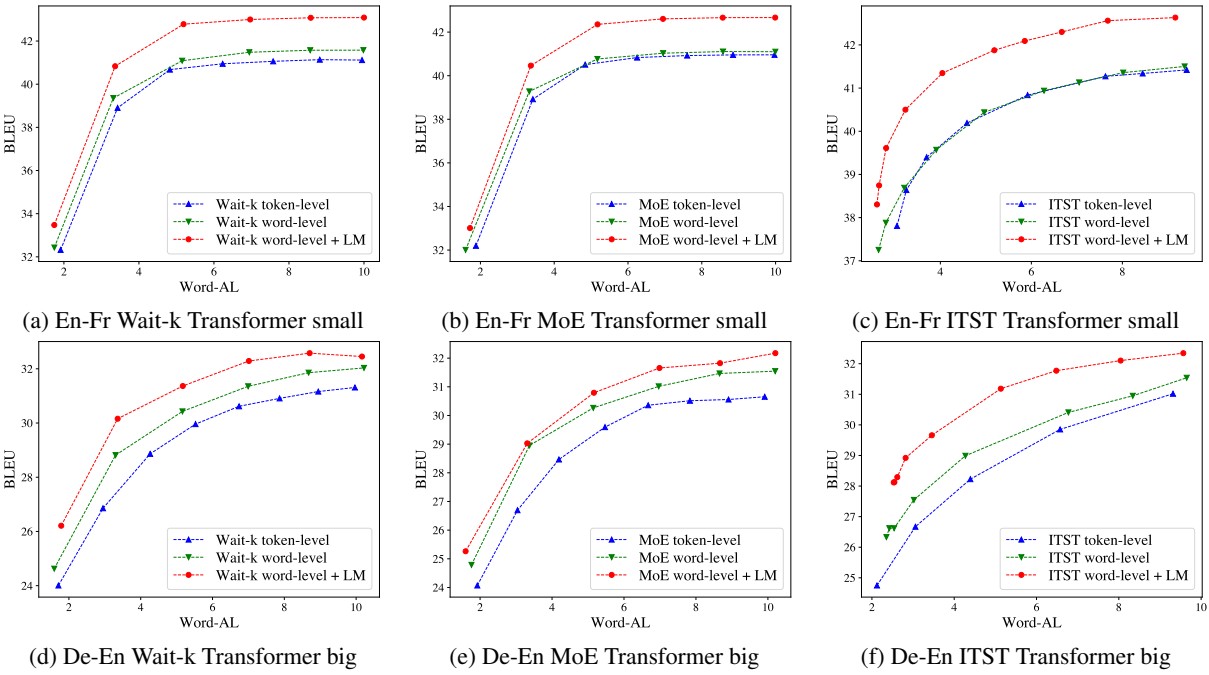

| | | |
|---|---|---|
| (a) En-Fr Wait-k Transformer small | (b) En-Fr MoE Transformer small | (c) En-Fr ITST Transformer small |
| (d) De-En Wait-k Transformer big | (e) De-En MoE Transformer big | (f) De-En ITST Transformer big |

Figure 5: Results of translation quality v.s. word-level AL.

## 4 Experiments

### 4.1 Datasets

We use the following two datasets for experiments: **IWSLT 17 English to French (En-Fr)** (Cettolo et al., 2017) consists of 242k pairs split into 233k training, 890 validation, and 8,597 test pairs. **WMT 15 German to English (De-En)** comprises 4.5 million pairs. We use newstest2013 (3,000 pairs) for validation and newstest2015 (2,169 pairs) for testing.

### 4.2 Experimental settings

Our methods are evaluated on the following systems, all of which are based on Transformer (Vaswani et al., 2017) with unidirectional encoding.

**Wait-k** (Ma et al., 2019a): A model operating under Wait-k policy. A single model is trained for all k values by randomly sampling k during training (Elbayad et al., 2020). For the word-level Wait-k policy, we define it as reading the first k words and then alternating between reading one source word and writing one target word. [3]

**MoE Wait-k** (Zhang and Feng, 2021): A Mixture-of-Experts model initially trained with fixed Experts weights and then fine-tuned with dynamic Experts weights. The word-level Wait-k policy of different k is applied to each expert.

**ITST** (Zhang and Feng, 2022b): A SoTA SiMT model equipped with Information-Transport-based policy that quantifies information weights from each source to the current target token. To implement word-level ITST, we convert the number of source tokens required for the first target token of each target word into the corresponding number of source words using Equation 2. Once the required number of source words is read, we complete the translation of the word. Additionally, we calculate the latency cost at the word level.

We compare each system by training both token-level and word-level models, with and without an LM. For models with an LM, we use XGLM-564M (Lin et al., 2022) and employ the two-stage training in which we first train the SiMT model without the LM and initialize the encoder and decoder from the LM-fused model with the pre-trained weights (Zhu et al., 2020). We also tested the single stage training where all trainable parameters are trained jointly with the LM from scratch. The difference of these strategies are discussed in Section 5.3. We tokenized and encoded the input using `sentencepiece` (Kudo and Richardson, 2018) and applied BPE with a vocabulary size of 32k. We use `sacreBLEU` for BLEU calculation (Post, 2018). For models with token-level policies, we trained

---

[3]Technically, the word-level policy derived from the token-level Wait-k through the conversion process in Section 3.3 can wait between 1 and $k$ tokens, depending on the input encoding. Therefore, it is not equivalent to the word-level Wait-k policy we define here, which always waits for k words.

models with the official implementations[4][5][6], and implemented word-level policies based on these implementations. More training details are described in A.2.

## 4.3 Main results

The performance of each system is compared in Figure 5. Notably, when measuring latency at the word level, the word-level policy proves to be highly effective for all three systems across different latency levels and datasets, resulting in superior performance. The only exception is observed in the En-Fr ITST models that demonstrate similar levels of performance. The incorporation of an LM using the proposed LM-fused attention further enhances the performance for all word-level configurations. This observation highlights the suitability of word-level policies for the LM-fused attention approach and underscores the effectiveness of leveraging an LM to enhance SiMT systems.

Notably, as depicted in Figure 11, the word-level policies also outperform or compete with the token-level policies in token-level latency. This can be attributed to the enhanced token representations under the word-level policy, thanks to the contextual information provided by all other tokens belonging to the same word for each token.

## 5 Analysis

To validate the effectiveness of word-level policies from multiple angles, we conduct several analyses on various settings. All the experiments were conducted on WMT De.En with `transformer-big` unless specified otherwise.

## 5.1 Ablation study

### 5.1.1 Effects of Word-level READ and WRITE

To gain insights into the functionality of word-level READ and WRITE actions, we trained Wait-k models with various policy settings and conducted a performance comparison. Specifically, we examined models with the following policy settings:

**WW**: word-level READ and WRITE.
**TW**: token-level READ and word-level WRITE.
**WT**: word-level READ and token-level WRITE.
**TkTk**: a simpler baseline policy which involves alternating reading k source tokens and writing k

[4]Efficient Wait-k: https://github.com/elbayadm/attn2d
[5]MoE Wait-k: https://github.com/ictnlp/MoE-Waitk
[6]ITST: https://github.com/ictnlp/ITST

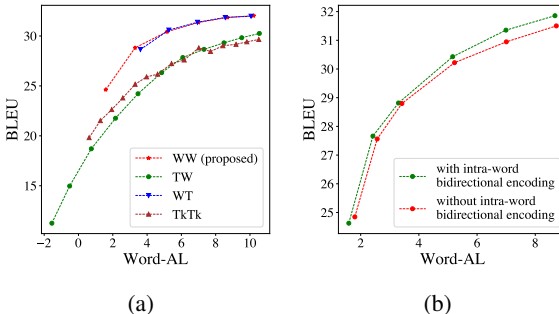

(a)                    (b)

Figure 6: Ablation studies for word-level policies. (a): Comparison of word-level Wait-k policies with different policies. (b): Comparison of word-level Wait-k models with and without the intra-word bidirectional encoding.

target tokens without considering word boundaries.

The results are presented in Figure 6 (a). The the word-level policy (**WW**) consistently outperforms **TW** across all latency settings. This is attributed to its imbalance between the number of source and target prefixes processed in each step. Additionally, **WT** achieves a minimal AL of approximately 3.6, indicating that it is not well-suited for scenarios that require low latency. Lastly, **TkTk** shows significantly worse performance than **WW**, suggesting that reading or writing a few consecutive tokens without considering semantic boundaries offers no benefits, unlike word-level policies.

### 5.1.2 Effects of intra-word bidirectional encoding

In order to assess the impact of the proposed intra-word bidirectional encoding, we trained word-level Wait-K models with and without it and compared the accuracy of the two models across different AL settings. The results are presented in Figure 6 (b).

Remarkably, the model equipped with the intra-word bidirectional encoding consistently achieved higher BLEU scores compared to the model without it, across all tested latency settings. This provides strong evidence of the effectiveness of the intra-word bidirectional encoding in enhancing SiMT performance.

## 5.2 Effectiveness of word-level policies for LM

In this subsection, we aim to explore the significance of word-level policies in leveraging LM for SiMT. We compare different configurations based on three factors:

**Word vs. Token**: The policy type that the model operates with.
**In-domain vocab vs. LM vocab**: Whether the

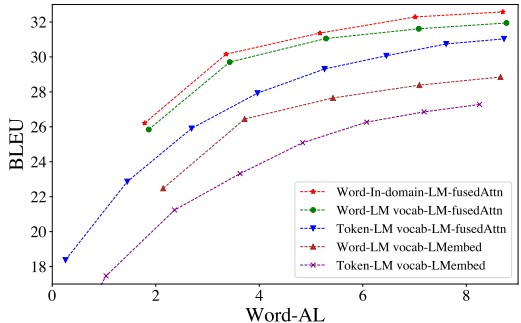

Figure 7: Comparison of models with different LM integration.

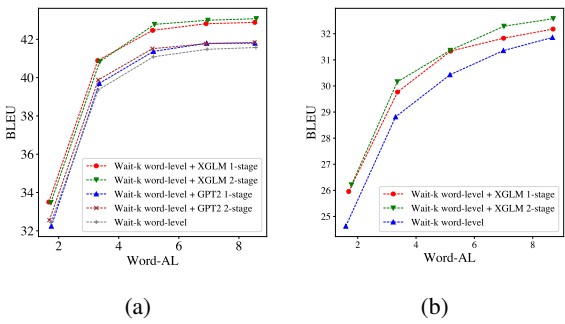

Figure 8: Translation accuracy comparison of LM-fused attention models with different training configurations. (a): En-Fr Transformer small (b) De-En Transformer big

model uses an in-domain vocabulary obtained from the in-domain training data or uses the LM's vocabulary for the source language. Note that the use of "in-domain vocab" is specific to the "Word" configuration due to the vocabulary mismatch.

**LM-fused attention vs. LM embed**: Whether the model incorporates an LM using the LM-fused attention or replacing the embedding layer of the encoder with the LM's embedding (Xu et al., 2021). The latter approach uses "LM vocab" by design.

Figure 7 showcases the results. The models with word-level policies consistently outperform those with token-level policies by a significant margin in both LM-fused attention and LM embedding settings, underscoring the importance of word-level policy adoption for effective LM integration. The top-performing configuration is the proposed **LMAttn-In-domain vocab-Word**, demonstrating that the highest translation accuracy is achieved when the SiMT model operates with an in-domain vocabulary. Additionally, it is evident that the "LM embed" approach performs notably worse than the proposed LM-fused attention, further affirming the latter's superiority.

### 5.3 Effects of LM-fused attention with various LMs and training configurations

To assess the effectiveness and broad applicability of our proposed LM integration, we conducted experiments on the IWSLT17 En-Fr dataset with two decoder-only LMs of different sizes: the 137M parameter GPT-2 model (Radford et al., 2018) and the XGLM-564M model. Additionally, we explore the option of training models in the single training stage instead of the two-stage training. The results, presented in Figure 8, demonstrate that GPT-2 model also exhibits improved performance with LM-fused attention, although their impact is

naturally less pronounced compared to XGLM due to the difference in model size. Moreover, although models trained using the single-stage training generally exhibit lower performance compared to those trained using the two-stage training, they still outperform models without the LM for most configurations. This indicates that the LM-fused attention is applicable to various types of LMs and remains effective even when using the single-stage training strategy. This flexibility allows users to choose a training approach and model configuration that aligns best with their desired accuracy goals and computational constraints.

### 5.4 Policy quality comparison

To assess the accuracy of policies in determining when to read or write, we adopt the methodology of estimating the quality of a policy in prior research (Zhang and Feng, 2022b,c; Guo et al., 2022). We measure the quality of a policy by analyzing the proportion of aligned source words received before translating on RWTH De-En alignment dataset [7].

To ensure accurate word-level alignment calculation, we consider an aligned source word is read before writing a ground truth (GT) target word if the last token of the source word is read before the first token of the target word is written. The results of this analysis are presented in Figure 9. It is observed that word-level policies, both for ITST and Wait-k, exhibit better alignments across most latency settings. This suggests that word-level policies contribute to revising premature WRITE actions by guiding the model to read the remaining tokens of the aligned word, without negatively impacting the model's latency.

---

[7] https://www-i6.informatik.rwth-aachen.de/goldAlignment/

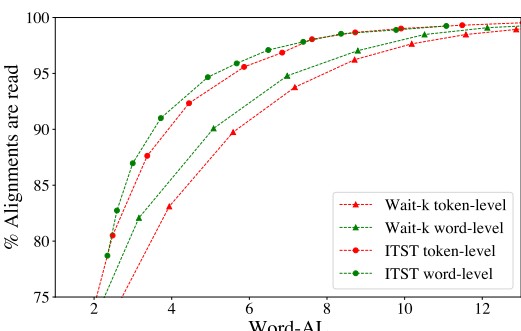

Figure 9: The percentage of aligned source words read before the translation of the target words started.

## 6 Conclusions

This paper explores the potential benefits of word-level operations in SiMT systems. We propose word-level latency calculation to ensure fair and accurate latency comparisons. We introduce a conversion process that transforms token-level policies into word-level policies, enabling the processing of multiple subwords that form a word within a single READ or WRITE action. Additionally, we propose the integration of LM-fused attention, which combines an autoregressive LM model into SiMT models with word-level policies. Experimental results demonstrate the superiority of word-level policies compared to token-level policies, as well as the effectiveness of the LM integration. Our findings highlight the crucial role of word-level policies in the integration process.

## 7 Limitations

While the proposed word-level policy implementation is widely applicable to most existing SiMT systems, it is important to note that systems utilizing languages with a writing style that lacks spaces or other delimiters between words or sentences (e.g., Chinese) are unable to derive benefits from this approach. Furthermore, it is important to consider that while the proposed LM-fused attention proves effective in enhancing translation quality across all latency levels, integrating a large LM may necessitate a faster compute capability to fulfill the low-latency demands of the SiMT task.

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

# A Appendix

## A.1 Token-level latency evaluation in previous works

Many previous works in the field of SiMT have employed token-level latency calculation as a fundamental metric for evaluating system performance. In the first group of papers, which includes (Arivazhagan et al., 2019; Ma et al., 2019b; Schneider and Waibel, 2020; Zheng et al., 2020; Zhang and Feng, 2022b), researchers explicitly stated their use of token-level latency calculation or incorporated token-level scores in their analyses. On the other hand, the second group, comprising (Zhang and Feng, 2022b; Elbayad et al., 2020; Zhang and Feng, 2022c, 2021, 2022a; Zhang et al., 2022; Guo et al., 2022; Zhang and Feng, 2023; Guo et al., 2023; Caglayan et al., 2020; Wang et al., 2023), can be identified by their publicly available official implementations, which incorporate token-level latency evaluation code. This underscores the prevalence of token-level latency calculation in assessing the effectiveness of SiMT systems in the cutting-edge SiMT research works.

## A.2 More training details

For Transformer model configurations, we follow the Efficient Wait-k (Elbayad et al., 2020)'s set-tings.[8] The details of hyperparameter settings for each model is in Table 1.

To ensure optimal performance of the ITST models, we conducted multiple training runs for each model configuration and dataset. We observed performance fluctuations across the training runs. To select the best-performing models for both token-level and word-level policies, we repeated the training process three times and selected the checkpoint with the lowest validation loss for each configuration.

For LM integrated models, the intra-word bidirectional encoding was not applied to the LM to prevent the need for fine-tuning.

## A.3 Examples and discussions

In Table 2, an illustrative case is presented to demonstrate the distinction between a token-level policy (token-level Wait-1) and a word-level policy (word-level Wait-1). At Step 2, the token-level model predicts the target token "B" after processing the subword "B" for the word "Beine" (which means "leg" in German). Subsequently, it fails to recover from this incorrect prediction and continues by predicting "ody___" as the next token, even after processing the remaining token "eine ___". In contrast, the word-level model accurately predicts "legs" (encoded as "leg s___") after processing the complete word 'Beine' (encoded as 'B eine___'). Furthermore, the token model also makes erroneous predictions for all steps when processing the word 'blutüberströmt.' (encoded as 'bl ut über ström t . ___', meaning "covered in blood" in German), while the word-level model accurately predicts a correct word in a single step.

Another case can be observed in Table 3, where the token-level model initiates the translation before fully reading the word "Irgendetwas" (encoded as "Irgen det was___"). As a consequence, it produces an incorrect translation. On the other hand, the word-level model accurately translates the sentence by processing the complete word before making any predictions.

## A.4 Table of main results

## A.5 Graphs of main results

---

[8]https://github.com/elbayadm/attn2d

| Hyperparameter | `transformer-small` | `transformer-base` | `transformer-big` |
|---|---|---|---|
| attention heads | 4 | 8 | 16 |
| embedding dime | 256 | 512 | 1024 |
| ffn embeding dim | 1024 | 2048 | 4096 |
| dropout | 0.3 | 0.3 | 0.3 |
| encoder layers | 6 | 6 | 6 |
| decoder layers | 6 | 6 | 6 |
| lr | 5e-4 | 5e-4 | 5e-4 |
| lr scheduler | inverse sqrt | inverse sqrt | inverse sqrt |
| optimizer | Adam (0.9, 0.98) | Adam (0.9, 0.98) | Adam (0.9, 0.98) |
| clip-norm | 0 | 0 | 0 |
| warmup-updates | 4000 | 4000 | 4000 |
| warmup-init-lr | 1e-7 | 1e-7 | 1e-7 |
| weight decay | 0 | 0 | 0 |
| label smoothing | 0.1 | 0.1 | 0.1 |
| max tokens | 32768 | 524288 | 1048576 |

Table 1: Hyperparameters of each Transformer model.

| **Src:** | Meine B eine waren bl ut über ström t . | | |
|---|---|---|---|
| **Ref:** | My leg s were covered in blood . | | |
| **Step** | **Streaminig Input** | **Token Wait-1 Output** | **Word Wait-1 Output** |
| 1 | Meine | My | My |
| 2 | Meine B | B | |
| 3 | Meine B eine | ody | leg s |
| 4 | Meine B eine waren | was | were |
| 5 | Meine B eine waren bl | blue | |
| 6 | Meine B eine waren bl ut | and | |
| 7 | Meine B eine waren bl ut über | my | |
| 8 | Meine B eine waren bl ut über ström | leg | |
| 9 | Meine B eine waren bl ut über ström t | s | |
| 10 | Meine B eine waren bl ut über ström t . | were blood - ri dden . | bloo dy . |

Table 2: An example case from WMT De.En test set (`transformer-big`) for Token-level Wait-1 (token-AL: 2.56, word-AL: 1.69) and Word-level Wait-1 (token-AL: 1.44, word-AL: 1.0).

| **Src:** | Ir gen det was lag in der Luft . | | |
|---|---|---|---|
| **Ref:** | Some thing was up . | | |
| **Step** | **Streaminig Input** | **Token Wait-1 Output** | **Word Wait-1 Output** |
| 1 | Ir | Ir | |
| 2 | Ir gen | gen | |
| 3 | Ir gen det | gen | |
| 4 | Ir gen det was | gen | Some thing |
| 5 | Ir gen det was lag | gen | lay |
| 6 | Ir gen det was lag in | s | in |
| 7 | Ir gen det was lag in der | , | the |
| 8 | Ir gen det was lag in der Luft | in | |
| 9 | Ir gen det was lag in der Luft . | fact , was in the air . | air . |

Table 3: An Eexample case from WMT De.En test set (`transformer-big`) for Token-level Wait-1 (token-AL: 2.69, word-AL: 4.25) and Word-level Wait-1 (token-AL: 2.38, word-AL: 1.0).

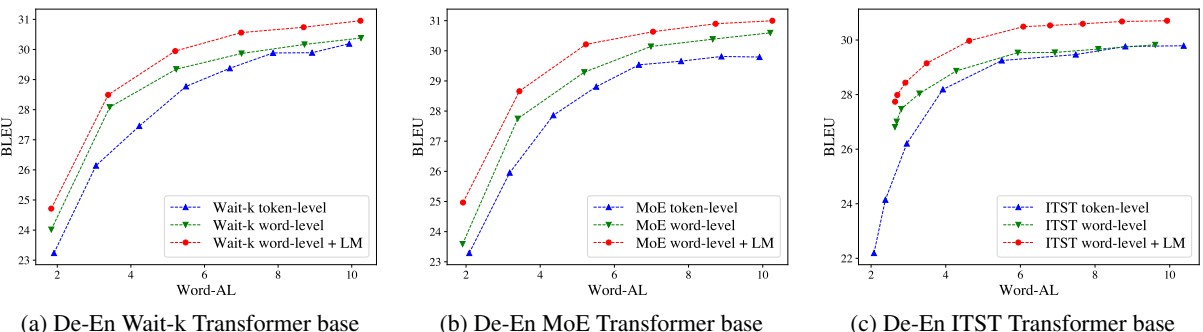

(a) De-En Wait-k Transformer base  (b) De-En MoE Transformer base  (c) De-En ITST Transformer base

Figure 10: Results of translation quality v.s. word-level AL, De-En `transformer-base`.

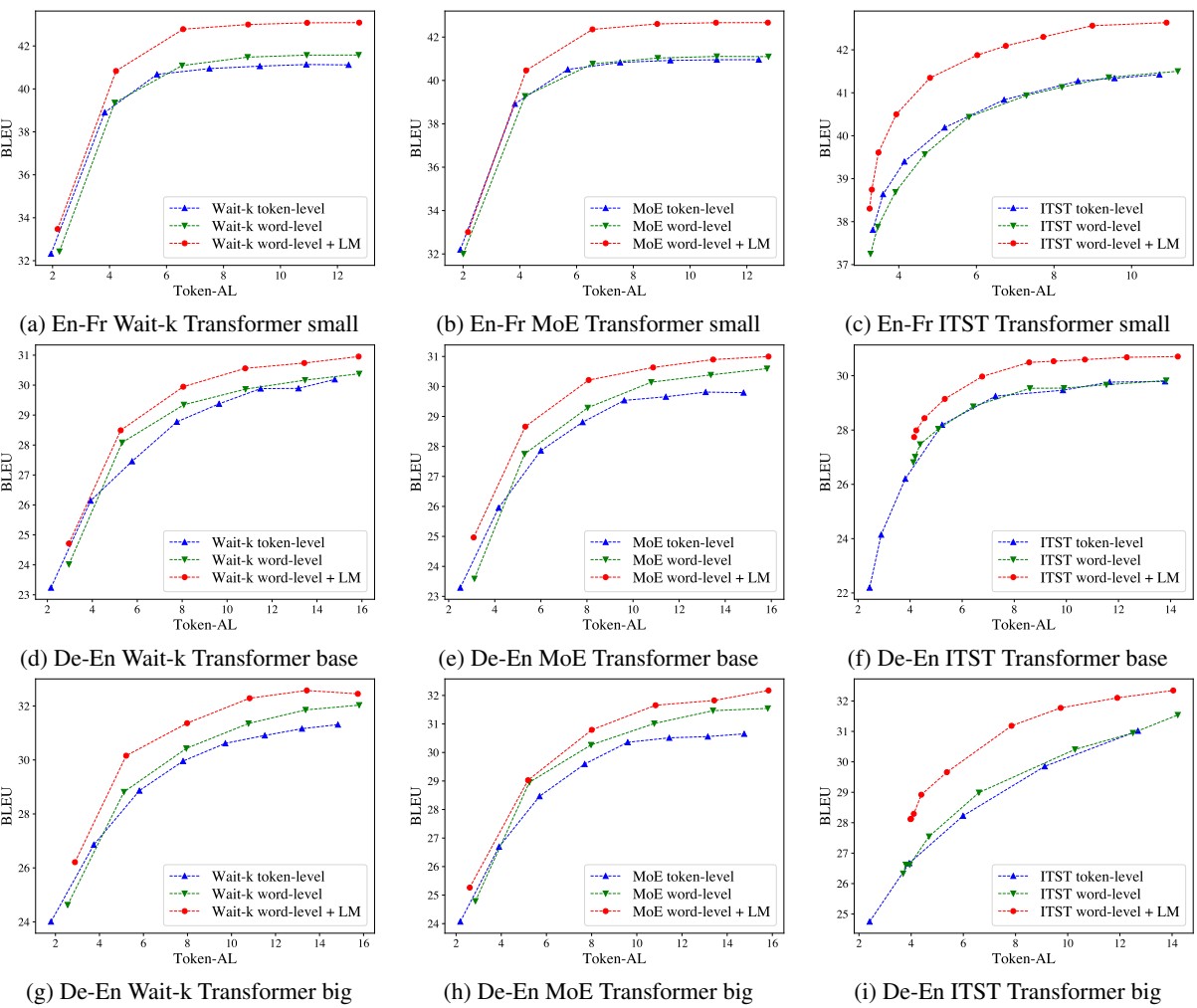

(a) En-Fr Wait-k Transformer small  (b) En-Fr MoE Transformer small  (c) En-Fr ITST Transformer small

(d) De-En Wait-k Transformer base  (e) De-En MoE Transformer base  (f) De-En ITST Transformer base

(g) De-En Wait-k Transformer big  (h) De-En MoE Transformer big  (i) De-En ITST Transformer big

Figure 11: Results of translation quality v.s. token-level AL.

| Token-level Wait-k | | | |
|---|---|---|---|
| k | token AL | word AL | BLEU |
| 1 | 1.95 | 1.91 | 32.32 |
| 3 | 3.83 | 3.43 | 38.90 |
| 5 | 5.67 | 4.82 | 40.68 |
| 7 | 7.51 | 6.23 | 40.95 |
| 9 | 9.28 | 7.58 | 41.06 |
| 11 | 10.91 | 8.82 | 41.14 |
| 13 | 12.39 | 9.95 | 41.12 |
| Word-level Wait-k | | | |
| k | token AL | word AL | BLEU |
| 1 | 2.24 | 1.74 | 32.43 |
| 3 | 4.18 | 3.32 | 39.36 |
| 5 | 6.55 | 5.15 | 41.09 |
| 7 | 8.85 | 6.94 | 41.48 |
| 9 | 10.92 | 8.57 | 41.57 |
| 11 | 12.75 | 9.98 | 41.58 |
| Word-level Wait-k w/ LM | | | |
| k | token AL | word AL | BLEU |
| 1 | 2.18 | 1.74 | 33.47 |
| 3 | 4.23 | 3.37 | 40.83 |
| 5 | 6.58 | 5.19 | 42.78 |
| 7 | 8.86 | 6.97 | 43.00 |
| 9 | 10.94 | 8.58 | 43.08 |
| 11 | 12.76 | 10.00 | 43.09 |

Table 4: Main results of `transformer-small` Waik-k models on IWLST17 En.Fr dataset.

| Token-level MoE Wait-k | | | |
|---|---|---|---|
| k | token AL | word AL | BLEU |
| 1 | 1.90 | 1.88 | 32.20 |
| 3 | 3.83 | 3.42 | 38.92 |
| 5 | 5.69 | 4.84 | 40.51 |
| 7 | 7.53 | 6.25 | 40.83 |
| 9 | 9.31 | 7.60 | 40.92 |
| 11 | 10.94 | 8.85 | 40.95 |
| 13 | 12.42 | 9.97 | 40.96 |
| Word-level MoE Wait-k | | | |
| k | token AL | word AL | BLEU |
| 1 | 2.01 | 1.60 | 32.01 |
| 3 | 4.18 | 3.32 | 39.28 |
| 5 | 6.57 | 5.17 | 40.77 |
| 7 | 8.86 | 6.95 | 41.03 |
| 9 | 10.94 | 8.58 | 41.11 |
| 11 | 12.76 | 9.99 | 41.10 |
| Word-level MoE Wait-k w/ LM | | | |
| k | token AL | word AL | BLEU |
| 1 | 2.17 | 1.72 | 33.01 |
| 3 | 4.22 | 3.37 | 40.46 |
| 5 | 6.56 | 5.17 | 42.35 |
| 7 | 8.84 | 6.95 | 42.61 |
| 9 | 10.91 | 8.57 | 42.66 |
| 11 | 12.74 | 9.99 | 42.67 |

Table 5: Main results of `transformer-small` MoE Waik-k models on IWSLT17 En.Fr dataset.

| Token-level ITST | | | |
|---|---|---|---|
| $\delta$ | token AL | word AL | BLEU |
| 0.2 | 3.33 | 3.05 | 37.81 |
| 0.3 | 3.59 | 3.25 | 38.64 |
| 0.4 | 4.14 | 3.70 | 39.40 |
| 0.5 | 5.18 | 4.59 | 40.19 |
| 0.6 | 6.71 | 5.92 | 40.84 |
| 0.7 | 8.62 | 7.63 | 41.27 |
| 0.75 | 9.56 | 8.45 | 41.34 |
| 0.8 | 10.72 | 9.41 | 41.42 |
| 0.85 | 12.11 | 10.53 | 41.42 |
| 0.9 | 13.84 | 11.80 | 41.42 |
| **Word-level ITST** | | | |
| $\delta$ | token AL | word AL | BLEU |
| 0.2 | 3.27 | 2.64 | 37.25 |
| 0.3 | 3.46 | 2.80 | 37.88 |
| 0.4 | 3.91 | 3.21 | 38.69 |
| 0.5 | 4.67 | 3.91 | 39.57 |
| 0.6 | 5.81 | 4.96 | 40.44 |
| 0.7 | 7.29 | 6.28 | 40.94 |
| 0.75 | 8.21 | 7.05 | 41.13 |
| 0.8 | 9.41 | 8.02 | 41.36 |
| 0.85 | 11.19 | 9.37 | 41.50 |
| 0.9 | 13.36 | 11.00 | 41.53 |
| **Word-level ITST w/ LM** | | | |
| $\delta$ | token AL | word AL | BLEU |
| 0.2 | 3.25 | 2.61 | 38.30 |
| 0.3 | 3.30 | 2.65 | 38.75 |
| 0.4 | 3.48 | 2.81 | 39.61 |
| 0.5 | 3.94 | 3.23 | 40.50 |
| 0.6 | 4.80 | 4.04 | 41.35 |
| 0.7 | 6.02 | 5.19 | 41.88 |
| 0.75 | 6.76 | 5.86 | 42.09 |
| 0.8 | 7.72 | 6.67 | 42.30 |
| 0.85 | 8.99 | 7.68 | 42.56 |
| 0.9 | 10.90 | 9.16 | 42.63 |

Table 6: Main results of `transformer-small` ITST models on IWSLT17 En.Fr dataset.

| Token-level Wait-k | | | |
|---|---|---|---|
| k | token AL | word AL | BLEU |
| 1 | 0.52 | 0.75 | 17.63 |
| 3 | 2.16 | 1.91 | 23.24 |
| 5 | 3.92 | 3.05 | 26.14 |
| 7 | 5.77 | 4.23 | 27.46 |
| 9 | 7.76 | 5.50 | 28.77 |
| 11 | 9.64 | 6.69 | 29.37 |
| 13 | 11.49 | 7.86 | 29.88 |
| 15 | 13.18 | 8.92 | 29.89 |
| 17 | 14.79 | 9.93 | 30.19 |
| **Word-level Wait-k** | | | |
| k | token AL | word AL | BLEU |
| 1 | 2.96 | 1.84 | 24.02 |
| 3 | 5.33 | 3.43 | 28.08 |
| 5 | 8.08 | 5.23 | 29.35 |
| 7 | 10.82 | 7.01 | 29.87 |
| 9 | 13.47 | 8.72 | 30.17 |
| 11 | 15.88 | 10.25 | 30.38 |
| **Word-level Wait-k w/ LM** | | | |
| k | token AL | word AL | BLEU |
| 1 | 2.96 | 1.84 | 24.71 |
| 3 | 5.27 | 3.38 | 28.49 |
| 5 | 8.05 | 5.20 | 29.95 |
| 7 | 10.81 | 7.00 | 30.56 |
| 9 | 13.44 | 8.69 | 30.74 |
| 11 | 15.85 | 10.24 | 30.95 |

Table 7: Main results of `transformer-base` Waik-k models on WMT15 De.En dataset.

| Token-level MoE Wait-k | | | |
| --- | --- | --- | --- |
| k | token AL | word AL | BLEU |
| 1 | 1.17 | 1.11 | 17.28 |
| 3 | 2.50 | 2.09 | 23.29 |
| 5 | 4.17 | 3.18 | 25.95 |
| 7 | 6.00 | 4.36 | 27.86 |
| 9 | 7.81 | 5.51 | 28.81 |
| 11 | 9.62 | 6.66 | 29.53 |
| 13 | 11.42 | 7.80 | 29.66 |
| 15 | 13.15 | 8.89 | 29.81 |
| 17 | 14.80 | 9.91 | 29.79 |
| Word-level MoE Wait-k | | | |
| k | token AL | word AL | BLEU |
| 1 | 3.12 | 1.91 | 23.59 |
| 3 | 5.29 | 3.39 | 27.75 |
| 5 | 8.03 | 5.19 | 29.29 |
| 7 | 10.79 | 6.98 | 30.14 |
| 9 | 13.38 | 8.65 | 30.39 |
| 11 | 15.81 | 10.2 | 30.6 |
| Word-level MoE Wait-k w/ LM | | | |
| k | token AL | word AL | BLEU |
| 1 | 3.08 | 1.92 | 24.97 |
| 3 | 5.32 | 3.44 | 28.66 |
| 5 | 8.08 | 5.23 | 30.21 |
| 7 | 10.87 | 7.04 | 30.63 |
| 9 | 13.47 | 8.73 | 30.9 |
| 11 | 15.88 | 10.26 | 31.0 |

Table 8: Main results of `transformer-base` MoE Waik-k models on WMT15 De.En dataset.

| Token-level ITST | | | |
| --- | --- | --- | --- |
| $\delta$ | token AL | word AL | BLEU |
| 0.2 | 2.45 | 2.08 | 22.19 |
| 0.3 | 2.89 | 2.38 | 24.15 |
| 0.4 | 3.82 | 2.96 | 26.21 |
| 0.5 | 5.23 | 3.92 | 28.19 |
| 0.6 | 7.28 | 5.50 | 29.25 |
| 0.7 | 9.87 | 7.48 | 29.47 |
| 0.75 | 11.66 | 8.81 | 29.76 |
| 0.8 | 13.79 | 10.38 | 29.79 |
| 0.85 | 16.60 | 12.11 | 29.90 |
| 0.9 | 19.81 | 13.85 | 29.71 |
| Word-level ITST | | | |
| $\delta$ | token AL | word AL | BLEU |
| 0.2 | 4.13 | 2.64 | 26.81 |
| 0.3 | 4.19 | 2.69 | 27.01 |
| 0.4 | 4.39 | 2.81 | 27.47 |
| 0.5 | 5.09 | 3.30 | 28.04 |
| 0.6 | 6.43 | 4.28 | 28.87 |
| 0.7 | 8.60 | 5.93 | 29.54 |
| 0.75 | 9.91 | 6.92 | 29.54 |
| 0.8 | 11.54 | 8.09 | 29.67 |
| 0.85 | 13.83 | 9.61 | 29.82 |
| 0.9 | 16.75 | 11.38 | 29.93 |
| Word-level ITST w/ LM | | | |
| $\delta$ | token AL | word AL | BLEU |
| 0.2 | 4.15 | 2.64 | 27.74 |
| 0.3 | 4.24 | 2.70 | 27.99 |
| 0.4 | 4.55 | 2.92 | 28.44 |
| 0.5 | 5.33 | 3.49 | 29.15 |
| 0.6 | 6.77 | 4.63 | 29.97 |
| 0.7 | 8.57 | 6.08 | 30.49 |
| 0.75 | 9.51 | 6.80 | 30.53 |
| 0.8 | 10.71 | 7.67 | 30.60 |
| 0.85 | 12.32 | 8.72 | 30.68 |
| 0.9 | 14.28 | 9.93 | 30.71 |

Table 9: Main results of `transformer-base` ITST models on WMT15 De.En dataset.

| Token-level Wait-k | | | |
|---|---|---|---|
| k | token AL | word AL | BLEU |
| 1 | -0.13 | 0.39 | 19.20 |
| 3 | 1.80 | 1.71 | 24.01 |
| 5 | 3.74 | 2.95 | 26.86 |
| 7 | 5.81 | 4.26 | 28.86 |
| 9 | 7.80 | 5.53 | 29.96 |
| 11 | 9.72 | 6.74 | 30.62 |
| 13 | 11.51 | 7.87 | 30.91 |
| 15 | 13.20 | 8.95 | 31.16 |
| 17 | 14.84 | 9.97 | 31.31 |
| Word-level Wait-k | | | |
| k | token AL | word AL | BLEU |
| 1 | 2.55 | 1.59 | 24.63 |
| 3 | 5.11 | 3.30 | 28.82 |
| 5 | 7.95 | 5.16 | 30.43 |
| 7 | 10.77 | 6.99 | 31.35 |
| 9 | 13.37 | 8.678 | 31.86 |
| 11 | 15.80 | 10.22 | 32.03 |
| Word-level Wait-k w/ LM | | | |
| k | token AL | word AL | BLEU |
| 1 | 2.89 | 1.79 | 26.21 |
| 3 | 5.21 | 3.36 | 30.16 |
| 5 | 7.98 | 5.17 | 31.36 |
| 7 | 10.82 | 7.01 | 32.29 |
| 9 | 13.42 | 8.71 | 32.58 |
| 11 | 15.74 | 10.16 | 32.45 |

Table 10: Main results of `transformer-big` Waik-k models on WMT15 De.En dataset.

| Token-level MoE Wait-k | | | |
|---|---|---|---|
| k | token AL | word AL | BLEU |
| 1 | 0.86 | 0.94 | 18.44 |
| 3 | 2.19 | 1.92 | 24.07 |
| 5 | 3.90 | 3.04 | 26.70 |
| 7 | 5.70 | 4.19 | 28.47 |
| 9 | 7.69 | 5.47 | 29.59 |
| 11 | 9.60 | 6.67 | 30.36 |
| 13 | 11.44 | 7.83 | 30.51 |
| 15 | 13.14 | 8.91 | 30.56 |
| 17 | 14.75 | 9.91 | 30.65 |
| Word-level MoE Wait-k | | | |
| k | token AL | word AL | BLEU |
| 1 | 2.86 | 1.76 | 24.79 |
| 3 | 5.24 | 3.37 | 28.95 |
| 5 | 7.98 | 5.15 | 30.26 |
| 7 | 10.77 | 6.97 | 31.01 |
| 9 | 13.37 | 8.66 | 31.47 |
| 11 | 15.80 | 10.20 | 31.55 |
| Word-level MoE Wait-k w/ LM | | | |
| k | token AL | word AL | BLEU |
| 1 | 2.60 | 1.60 | 25.26 |
| 3 | 5.19 | 3.31 | 29.03 |
| 5 | 8.01 | 5.16 | 30.79 |
| 7 | 10.83 | 6.99 | 31.66 |
| 9 | 13.43 | 8.67 | 31.82 |
| 11 | 15.83 | 10.21 | 32.17 |

Table 11: Main results of `transformer-big` MoE Waik-k models on WMT15 De.En dataset.

| Token-level ITST | | | |
|---|---|---|---|
| $\delta$ | token AL | word AL | BLEU |
| 0.2 | 1.49 | 1.55 | 23.22 |
| 0.3 | 2.41 | 2.12 | 24.75 |
| 0.4 | 3.93 | 3.06 | 26.67 |
| 0.5 | 5.99 | 4.39 | 28.23 |
| 0.6 | 9.12 | 6.57 | 29.85 |
| 0.7 | 12.69 | 9.32 | 31.02 |
| 0.75 | 14.72 | 10.75 | 31.28 |
| 0.8 | 16.95 | 12.23 | 31.54 |
| 0.85 | 19.06 | 13.49 | 31.70 |
| 0.9 | 21.26 | 14.69 | 31.84 |
| Word-level ITST | | | |
| $\delta$ | token AL | word AL | BLEU |
| 0.2 | 3.69 | 2.35 | 26.33 |
| 0.3 | 3.79 | 2.42 | 26.62 |
| 0.4 | 3.94 | 2.54 | 26.62 |
| 0.5 | 4.69 | 3.02 | 27.55 |
| 0.6 | 6.60 | 4.27 | 28.99 |
| 0.7 | 10.28 | 6.77 | 30.41 |
| 0.75 | 12.50 | 8.34 | 30.95 |
| 0.8 | 14.22 | 9.65 | 31.54 |
| 0.85 | 15.90 | 10.81 | 31.56 |
| 0.9 | 18.13 | 12.23 | 31.61 |
| Word-level ITST w/ LM | | | |
| $\delta$ | token AL | word AL | BLEU |
| 0.2 | 3.97 | 2.53 | 28.12 |
| 0.3 | 4.00 | 2.55 | 28.13 |
| 0.4 | 4.10 | 2.62 | 28.29 |
| 0.5 | 4.39 | 2.82 | 28.92 |
| 0.6 | 5.37 | 3.46 | 29.66 |
| 0.7 | 7.85 | 5.13 | 31.18 |
| 0.75 | 9.73 | 6.48 | 31.77 |
| 0.8 | 11.90 | 8.04 | 32.10 |
| 0.85 | 14.04 | 9.57 | 32.34 |
| 0.9 | 16.14 | 10.97 | 32.62 |

Table 12: Main results of `transformer-big` ITST models on WMT15 De.En dataset.