# OpenReview forum: "Enhanced Simultaneous Machine Translation with Word-level Policies"
_EMNLP/2023/Conference — EMNLP 2023 Findings_

### Official Review · Reviewer_5oLi · 2023-08-04

**Soundness:** 3

**Excitement:**

3: Ambivalent: It has merits (e.g., it reports state-of-the-art results, the idea is nice), but there are key weaknesses (e.g., it describes incremental work), and it can significantly benefit from another round of revision. However, I won't object to accepting it if my co-reviewers champion it.

**Justification For Ethical Concerns:**

From Line 42: "The performance analysis and implementation of most SiMT systems, to the best of our knowledge, have been carried out on encoded sequences of source and target subwords, rather than on the original source and target sentences. This has led to two critical issues that need to be addressed."

This is simply not true! And because of this, the paper has no value! The authors did add "to the best of our knowledge", but actually, this is a known fact for everyone working on simultaneous MT.

=== After the author's rebuttal, I acknowledge that some papers still use BPE-level evaluation of simultaneous MT systems. Still, I believe that it is clear that this is not the right thing to do - yet I don't have any ethical concerns anymore after the authors re-worded the paper.

**Missing References:**

@inproceedings{papi2022does, title={Does Simultaneous Speech Translation need Simultaneous Models?}, author={Papi, Sara and Gaido, Marco and Negri, Matteo and Turchi, Marco}, booktitle={Findings of the Association for Computational Linguistics: EMNLP 2022}, pages={141--153}, year={2022} }

@article{wilken2020neural, title={Neural Simultaneous Speech Translation Using Alignment-Based Chunking}, author={Wilken, Patrick and Alkhouli, Tamer and Matusov, Evgeny and Golik, Pavel}, journal={IWSLT 2020}, pages={237}, year={2020} }

**Paper Topic And Main Contributions:**

The paper proposes to use policies like wait-k for simultaneous MT on word level rather than on subword level. The authors additionally suggest to use a language model (possibly with a different subword segmentation than the translation model) to further improve the quality of the translation; operating on word level makes it possible to apply the LM despite the differences in subword segmentation.

**Reasons To Accept:**

One of the main claims of the paper that so far, the average lagging, one of the main ways of evaluating simultaneous MT systems, has happened on subword level. This claim is not true, and the idea to go from subword level to word level when computing the policy is not new at all either. The remaining part about using the LM is not enough for publication.

So overall, I see no reasons to accept the paper in its current form.

**Reasons To Reject:**

The main claim of the paper that so far the average lagging was computed on subword level and not on word level is simply not true. In fact, SimEval by default uses word level.  This is absolutely necessary to do to have fair comparison between submissions to competitive evaluations/shared tasks like Workshop on Simultaneous Translation, and also Simultaneous speech translation track at the IWSLT conference.

Also, prior research definitely used word-level adaptive policies, like the phrases/chunks in one of the papers that I listed below - these were defined on the word level.

Your use of word-level "chunk attention" is misleading - you seem to just attend to multiple subwords of the same word - this is not a chunk in any of the established meaning of this word.

Overall, it is clear that the output of words, not subwords is expected in simultaneous MT, as they have to be presented to a user in form of words. How you read the input, in subwords, words, or directly from speech, or in characters - this is up to the architecture of the system, so there is nothing novel about going from subwords to words there.

So overall, I see no merit in the paper, as its main claims have no ground.

**Reproducibility:**

3: Could reproduce the results with some difficulty. The settings of parameters are underspecified or subjectively determined; the training/evaluation data are not widely available.

**Reviewer Confidence:**

4: Quite sure. I tried to check the important points carefully. It's unlikely, though conceivable, that I missed something that should affect my ratings.

**Typos Grammar Style And Presentation Improvements:**

Line 493: "all tested latency settings": according to Fig. 6b, this is not true: in the lowest latency setting of 2, the system without "chunk attention" is better.

---

> ### Author Rebuttal · Authors · 2023-08-28
>
> Thank you for your review, comments and opinions.
>
> ### In-depth explanation and evidence supporting our claim regarding the evaluation and analysis of latency
> First of all, to clarify, our claim pertains specifically to SiMT for text-to-text translation, where both source and target sentences are typically BPE tokenized. We have updated our paper to make this distinction clear. \
> While competitions ensure a fair comparison by defining a standardized input/output format that all submissions must adhere to, many individual research works report latency scores without such a standardized format. We did not claim that word-level latency calculations were never used. However, as shown in the below references, token-level AL has been more common in the recent papers and we proposed to use word-level AL for a fair comparison. We think this situation is analogous to the inconsistent BLEU calculation before the introduction of SacreBLEU, which was affected by variations in tokenization and normalization. Below we summarize a list of references that we have reviewed to confirm that the latency evaluations of the systems were conducted on encoded tokens:
> 1. The MILk paper [1] is known to use BPE tokens for latency calculation. Examples and analysis in the paper are also based on tokens.
> 2. In the MMA paper [2], a footnote in page 6 mentions they follow [1]: “Latency metrics are computed on BPE tokens for WMT15 De-En – consistent with Arivazhagan et al.”
> 3. [3] also states that they report latency scores on tokenized inputs following [1].
> 4. In [4], the example sentences and their latency scores are at the token level (page 5).
> 5. In [5], all latency scores for text inputs are all measured at the token level.
> 6.  [5, 6, 7, 8, 9, 10, 11, 12, 13, 14] are SiMT research works that we are aware of that share official implementations publicly. We analyzed and ran all the code, and confirmed that they all calculated the latency score from the subword sequences.
> 7. We asked the main author of [15] how they calculated the AL scores and this is a quote from their reply: “Our latency(AL) scores were calculated on BPE tokens. We generally followed the training/ evaluation settings of  Multipath Wait-k and ITST, and BPE-level Average Lagging (AL) were used to measure latency.”
> 8. SimulEval supports two types of “--eval-latency-unit”: “word” or “char”, and the default is “word”. However, the code relies on the user to input original source/target text and the given agent to calculate a word-level latency. If BPE tokens are given, it may return BPE-token level latency scores. More importantly, all the open-source implementations we mentioned above do not use SimulEval and instead use their own evaluation code.
>
> We believe that the above evidence could advocate our claim and thus we respectfully request the reviewer to consider reevaluating our paper. We have added the above details to the appendix in the paper. We also revised the words "most SiMT systems" in our claim line 42 to "many SiMT systems".
>
> [1] Monotonic Infinite Lookback Attention for Simultaneous Machine Translation (Arivazhagan et al., ACL 2019) \
> [2] Monotonic Multihead Attention (Ma et al., ICLR 2020) \
> [3] Towards Stream Translation: Adaptive Computation Time for Simultaneous Machine Translation (Schneider & Waibel, IWSLT 2020) \
> [4] Simultaneous Translation Policies: From Fixed to Adaptive (Zheng et al., ACL 2020) \
> [5] Information-Transport-based Policy for Simultaneous Translation (Zhang & Feng, EMNLP 2022) \
> [6] Efficient Wait-k Models for Simultaneous Machine Translation (Elbayad et al., INTERSPEECH 2020)  \
> [7] Modeling Dual Read/Write Paths for Simultaneous Machine Translation (Zhang & Feng, ACL 2022) \
> [8] Universal Simultaneous Machine Translation with Mixture-of-Experts Wait-k Policy (Zhang & Feng, EMNLP 2021) \
> [9] Gaussian Multi-head Attention for Simultaneous Machine Translation (Zhang & Feng, Findings 2022) \
> [10] Wait-info Policy: Balancing Source and Target at Information Level for Simultaneous Machine Translation (Zhang et al., Findings 2022) \
> [11] Turning Fixed to Adaptive: Integrating Post-Evaluation into Simultaneous Machine Translation (Guo et al., Findings 2022) \
> [12] Hidden Markov Transformer for Simultaneous Machine Translation (Zhang et al., ICLR 2023) \
> [13] Learning Optimal Policy for Simultaneous Machine Translation via Binary Search (Guo et al., ACL 2023) \
> [14] Simultaneous Machine Translation with Visual Context (Caglayan et al., EMNLP 2020) \
> [15] Better Simultaneous Translation with Monotonic Knowledge Distillation (Wang et al., ACL 2023)
>
> ### Existence of prior research using word-level policies and novelty of our work
> We acknowledge the second reference provided by the reviewer, which introduces a component for determining chunk boundaries at the word level. This work is indeed relevant and will be included in our Related Work section. However, it's important to emphasize that our contribution lies not in being the first to introduce word-level policies but in our discovery that many existing policies, initially designed at the subword level, can benefit significantly from transitioning to the word level. Moreover, we demonstrate that incorporating a language model on top of these word-level policies further enhances their performance. This finding represents a significant contribution to the SiMT community, as it opens the door for improving existing systems with relatively minor modifications and highlights the pivotal role of word-level policies in effectively integrating language models into SiMT.
>
> ### Naming issue with the “word-level chunk attention”
> We agree with you and renamed it to **intra-word bidirectional encoding**. Please refer to our rebuttal for Reviewer 2 if you need more detailed explanation.
>
> ### About the ethical concern of the reviewer that the authors are lying
> We are not lying about anything. We hope the evidence we provided above could resolve both the reviewer’s technical and ethical concerns.

---

### Official Review · Reviewer_CvhJ · 2023-08-04

**Soundness:** 4

**Excitement:**

4: Strong: This paper deepens the understanding of some phenomenon or lowers the barriers to an existing research direction.

**Missing References:**

When discussing the encoding of the input sequence in lines 227-232, I think it is worth mentioning two additional references:

@inproceedings{iranzo-sanchez-etal-2022-simultaneous,
    title = "From Simultaneous to Streaming Machine Translation by Leveraging Streaming History",
    author = "Iranzo Sanchez, Javier  and
      Civera, Jorge  and
      Juan-C{\'\i}scar, Alfons",
    booktitle = "Proceedings of the 60th Annual Meeting of the Association for Computational Linguistics (Volume 1: Long Papers)",
    month = may,
    year = "2022",
    address = "Dublin, Ireland",
    publisher = "Association for Computational Linguistics",
    url = "https://aclanthology.org/2022.acl-long.480",
    doi = "10.18653/v1/2022.acl-long.480",
    pages = "6972--6985",
    abstract = "Simultaneous machine translation has recently gained traction thanks to significant quality improvements and the advent of streaming applications. Simultaneous translation systems need to find a trade-off between translation quality and response time, and with this purpose multiple latency measures have been proposed. However, latency evaluations for simultaneous translation are estimated at the sentence level, not taking into account the sequential nature of a streaming scenario. Indeed, these sentence-level latency measures are not well suited for continuous stream translation, resulting in figures that are not coherent with the simultaneous translation policy of the system being assessed. This work proposes a stream-level adaptation of the current latency measures based on a re-segmentation approach applied to the output translation, that is successfully evaluated on streaming conditions for a reference IWSLT task",
}

@inproceedings{kahardipraja-etal-2021-towards,
    title = "Towards Incremental Transformers: An Empirical Analysis of Transformer Models for Incremental {NLU}",
    author = "Kahardipraja, Patrick  and
      Madureira, Brielen  and
      Schlangen, David",
    booktitle = "Proceedings of the 2021 Conference on Empirical Methods in Natural Language Processing",
    month = nov,
    year = "2021",
    address = "Online and Punta Cana, Dominican Republic",
    publisher = "Association for Computational Linguistics",
    url = "https://aclanthology.org/2021.emnlp-main.90",
    doi = "10.18653/v1/2021.emnlp-main.90",
    pages = "1178--1189",
    abstract = "Incremental processing allows interactive systems to respond based on partial inputs, which is a desirable property e.g. in dialogue agents. The currently popular Transformer architecture inherently processes sequences as a whole, abstracting away the notion of time. Recent work attempts to apply Transformers incrementally via restart-incrementality by repeatedly feeding, to an unchanged model, increasingly longer input prefixes to produce partial outputs. However, this approach is computationally costly and does not scale efficiently for long sequences. In parallel, we witness efforts to make Transformers more efficient, e.g. the Linear Transformer (LT) with a recurrence mechanism. In this work, we examine the feasibility of LT for incremental NLU in English. Our results show that the recurrent LT model has better incremental performance and faster inference speed compared to the standard Transformer and LT with restart-incrementality, at the cost of part of the non-incremental (full sequence) quality. We show that the performance drop can be mitigated by training the model to wait for right context before committing to an output and that training with input prefixes is beneficial for delivering correct partial outputs.",
}

**Paper Topic And Main Contributions:**

This paper shows that moving from token-based to word-level policies in Simultaneous Machine Translation (SiMT) brings about improvements in translation quality leveraging all subwords that are part of the word being read and not writing in the middle of a word. In addition, and more importantly, it eases the integration of external LMs in order to improve SiMT models, and this is shown applying LM-fused attention previously applied in NMT. Finally, word-level policies allows for a fair comparative evaluation independently of the subword algorithm being applied. The results show a consistent improvements up to approximately 15%.

**Questions For The Authors:**

Question A: How are the chunk boundaries defined? what are the parameters governing the model that takes the splitting decision? Obviously, the chunk size affects the quality of the bidirectional encoding, have you explored this point in your experiments? I guess bidirectional encoding is applied to all chunks, except for the last one in which unidirectional encoding would be applied, am I right?

**Reasons To Accept:**

The improvements in translation quality are consistent and significant across k values in wait-k policies.

**Reasons To Reject:**

The presentation and discussion of results need to be improved to put into perspective what contributions are having a more significant impact than others and under which conditions. For example, Figure 5 shows too many plots and too many curves, I would discard Transformer base plots and I would select those curves that clearly show the main conclusions that can be extracted from the experiments: word-level policies improve the results and adding a LM does much more. Those conditions that do not help much can be mentioned in the discussion. However, it would be nice to know what the contribution is for the word-level policies and the LMs and under which conditions one may contribute more than the other, if this is the case.

Word-level chunk attention is not clear to me since little detail is provided about how chunks are defined.

To sum up, the results are good and extensive, but are not easy to read in order to draw the main conclusions. The authors should have devoted more time to prepare the presentation of the results to ease the task of the reader.

**Reproducibility:**

4: Could mostly reproduce the results, but there may be some variation because of sample variance or minor variations in their interpretation of the protocol or method.

**Reviewer Confidence:**

3: Pretty sure, but there's a chance I missed something. Although I have a good feel for this area in general, I did not carefully check the paper's details, e.g., the math, experimental design, or novelty.

**Typos Grammar Style And Presentation Improvements:**

Typos:

L. 142: attetnion -> attention
L. 216: , The goal -> , the goal
L. 232: unidirectionally**.** (Elbayad et al., 2020)
L. 147: missing words in "Zhang and Feng (2022a) model to predict the alignment..."
L.422: BLEU calculation**.** (Post, 2018)
Table 2: An Eexample case -> An example case


Presentation improvements:

Tables 4-12 in the appendix could be more compacted and elaborated to ease the comparison across k values (since explored k values are different), relative improvements over the baseline would help to clearly see the impact of the different factors.

Figure 6 (a) and its description in the text need to be improved to make the description more consistent, you are using WW, TW, WT, but Read-k-Write-k to refer to TT policy.

I would discard Figure 6 (b), since it can be described with a sentence. In addition, this contribution is not clear to me since little detail is provided about how chunks are defined.

Figure 8 (a) contains too many curves that cannot be read properly. I would reduce the number of curves discarding those less significant in terms of impact in the results and describe these minor effects in the text. In addition, from an aesthetic viewpoint, I would try to use same font size across figures in page8 and shorthen legend description to not occlude the curves (see Figure 8 (a)).

---

> ### Author Rebuttal · Authors · 2023-08-28
>
> Thank you for your thorough review and constructive criticism. We have revised our paper to address your suggestions and comments. More details are elaborated below.
>
> ### Improving the presentation and discussion of results
> In the main result (Figure 5), We relocated the Transformer-base plot and token-level AL plots to the Appendix, and now have dedicated plots for each policy:  Wait-k, MoE, and ITST. With these changes, the total number of plots remains at 6 (3 policies * 2 datasets), but each plot features only 3 curves of baseline, word-level policies with and without an LM. Now it is a lot easier to identify the relative impact of word-level policies and adding an LM to them. (In short, both almost always improve results, but in general adding an LM bring more improvements)
>
> ### Explanation for the word-level chunk attention
> We acknowledge the concerns raised by two reviewers regarding the misleading naming. We have now renamed it **'intra-word bidirectional encoding.'** At the word level, this approach involves unidirectional encoding for each word in the input sentence, meaning past words cannot attend to future words. However, at the subword level, subwords within the same word can attend to each other, allowing past subwords to attend to future subwords within the same word. Since READ operates at the word level in word-level policies, this encoding does not require recomputation during each WRITE operation. It only necessitates a single forward pass, similar to token-level unidirectional encoding. However, it can produce a better encoded representation by enabling attention to more tokens.
>
> ### Missing references on discussing the encoding of the input sequence
> We have added these to the discussion in the paper like the following: "To minimize redundancy in encoding input sequences following READ operations, one can consider adaptations to the encoder for enhanced efficiency. Techniques such as the recurrent Linear Transformer (Kahardipraja et al., 2021), Partial Bidirectional Encoding (Sanchez et al., 2022), and the widely adopted unidirectional token encoding (Elbayad et al., 2020) are among the available options."
>
> ### Tables 4-12 in the appendix
> These tables present precise values corresponding to the main results depicted in Figure 5. Given that SiMT evaluation predominantly entails graphical comparisons of latency-accuracy curves, including these tables allows a more accurate comparison and recreating the same curves. Demonstrating relative improvements over the baseline in these tables poses challenges due to the varying latency (AL) scores associated with each data point. JFYI, we adopted the same table format other papers have used.
>
> ### WW, TW, WT, but Read-k-Write-k in Figure 6 (a)
> We chose not to use 'TT' for Read-k-Write-k to prevent confusion. According to our existing nomenclature of WW, TW, and WT, 'TT' should represent the token-level Wait-k, which is distinct from Read-k-Write-k. We acknowledge the need for a more consistent and reader-friendly naming scheme and have decided to change it to 'TkTk'.
>
> ### Cleaning Figure 8 (a)
> Following your advice, we reduced the number of curves from 7 to 5 in Fig. 8 (a) by removing the GPT2-medium results and putting them in the appendix.

---

### Official Review · Reviewer_tsh7 · 2023-08-06

**Soundness:** 4

**Excitement:**

4: Strong: This paper deepens the understanding of some phenomenon or lowers the barriers to an existing research direction.

**Paper Topic And Main Contributions:**

The paper addresses the problem of the assumption in many existing studies that operations are carried out at the subword level, even though the standard unit for input and output in most practical scenarios is typically at the word level. The paper demonstrates that policies devised and validated at the subword level are surpassed by those operating at the word level, which process multiple subwords to form a complete word in a single step. The main contributions of this paper are the demonstration of the superiority of word-level policies over subword-level policies in SiMT and the proposal of a method to boost SiMT models using language models.

**Questions For The Authors:**

1. What are some potential solutions or alternative approaches that could be explored to address the limitation of the proposed method in SiMT systems for languages with a writing style that lacks spaces or other delimiters between words or sentences, such as Chinese?

**Reasons To Accept:**

Strengths of the paper:
- Proposes a novel method of integrating language models (LM) into Simultaneous Machine Translation (SiMT) systems
- Offers a more versatile solution that can be integrated into most existing neural SiMT models
- LM-fused attention proves effective in enhancing translation quality across all latency levels
- Proposed word-level policy plays a crucial role in effectively managing the vocabulary mismatch between the LM and model

**Reasons To Reject:**

Weaknesses of the paper:
- The proposed method may not be applicable to languages with a writing style that lacks spaces or other delimiters between words or sentences, such as Chinese. Experiments like translation direction of En-Zh should be included in this paper.
- Integrating a large LM may require a faster compute capability to fulfill the low-latency demands of the SiMT task.Speed experiments should be involved in this paper.

**Reproducibility:**

4: Could mostly reproduce the results, but there may be some variation because of sample variance or minor variations in their interpretation of the protocol or method.

**Reviewer Confidence:**

4: Quite sure. I tried to check the important points carefully. It's unlikely, though conceivable, that I missed something that should affect my ratings.

---

> ### Author Rebuttal · Authors · 2023-08-28
>
> Thank you for your positive feedback and valuable comments.
>
> ### Speed experiments
> Following the reviewer's suggestion, we have added speed comparison between models with and without an LM to assess the potential impact of the LM integration to the latency of an SiMT system. Below is the results (Measured on a 3090 GPU):
>
> | Model | Speed (Words/sec)  |  Relative speed w.r.t. baseline|
> |:---:|:---:|:---:|
> | Transformer-small  | 72.0   | 1.0  |
> | Transformer-small + GPT2  | 47.6   | 0.66  |
> | Transformer-big  | 63.2   | 1.0  |
> | Transformer-big + XGLM  | 43.8  | 0.69  |
>
> Results show that models with an LM is up to 34% slower than baseline models. However, they are still faster than the speed of natural conversation (160 words per min) [1].
>
> ### Potential solutions or alternative approaches for languages without space
> For languages that have well-developed segmentation methods like 'jieba' for Chinese and 'nagisa' for Japanese, we can initially tokenize input sentences into words using the segmenter and then apply BPE to further split them into subwords, similar to [2]. However, a limitation that remains in this approach is that, it requires different systems to use the same segmenter for fair comparisons between them.
>
> [1] Towards an integrated understanding of speaking rate in conversation (Yuan et al., ICSLP 2006) \
> [2] The University of Edinburgh’s Neural MT Systems for WMT17 (Sennrich et al., WMT 2017)

---

### Meta-Review · Area_Chair_pXMF · 2023-09-07

**Recommendation:** 5

**Metareview:**

The main contributions of this paper are porting subword-level policies for Simultaneous Machine Translation to the word level and proposing fusion with a language model.

Altogether the reviewers agree that the paper is technically sound. There are some concerns about the presentation, and one reviewer believes that it will have limited impact due to the main idea being trivial.
However, given that (some) recent previous work has failed to explore this path, I believe that this only strengthens the merit of publishing the paper.

---

### Decision · Program_Chairs · 2023-10-07

**Decision:**

Accept-Findings

**Comment:**

The main contributions of this paper are porting subword-level policies for Simultaneous Machine Translation to the word level and proposing fusion with a language model.

Altogether the reviewers agree that the paper is technically sound. There are some concerns about the presentation, and one reviewer believes that it will have limited impact due to the main idea being trivial.
However, given that (some) recent previous work has failed to explore this path, I believe that this only strengthens the merit of publishing the paper.